# Dietary Diversity and Nutritional Adequacy among an Older Spanish Population with Metabolic Syndrome in the PREDIMED-Plus Study: A Cross-Sectional Analysis

**DOI:** 10.3390/nu11050958

**Published:** 2019-04-26

**Authors:** Naomi Cano-Ibáñez, Alfredo Gea, Miguel A. Martínez-González, Jordi Salas-Salvadó, Dolores Corella, M. Dolors Zomeño, Dora Romaguera, Jesús Vioque, Fernando Aros, Julia Wärnberg, J. Alfredo Martínez, Lluis Serra-Majem, Ramón Estruch, Francisco J. Tinahones, José Lapetra, Xavier Pintó, Josep A. Tur, Antonio García-Ríos, Blanca Riquelme-Gallego, Miguel Delgado-Rodríguez, Pilar Matía, Lidia Daimiel, Vicente Martín, Josep Vidal, Clotilde Vázquez, Emilio Ros, Pilar Buil-Cosiales, Andrés Díaz-López, Rebeca Fernández-Carrión, Montserrat Fitó, Jadwiga Konieczna, Leyre Notario-Barandiaran, Ángel M. Alonso-Gómez, Eugenio Contreras-Fernández, Itziar Abete, Almudena Sánchez-Villegas, Rosa Casas, Araceli Muñoz-Garach, José Manuel Santos-Lozano, Laura Gallardo-Alfaro, Josep Basora, Olga Portoles, Miguel Ángel Muñoz, Manuel Moñino, Salvador Miralles Gisbert, Anai Moreno Rodríguez, Miguel Ruiz-Canela, Antoni Palau Galindo, Karla Alejandra Pérez-Vega, Aurora Bueno-Cavanillas

**Affiliations:** 1Department of Preventive Medicine and Public Health, University of Granada, 18011 Granada, Spain; ncaiba@ugr.es (N.C.-I.); blanca.riquel@gmail.com (B.R.-G.); 2CIBER Epidemiología y Salud Pública (CIBERESP). Instituto de Salud Carlos III (ISCIII), 28029 Madrid, Spain; vioque@umh.es (J.V.); mdelgado@ujaen.es (M.D.-R.); vicente.martin@unileon.es (V.M.); lnotario@umh.es (L.N.-B.); sjmirallesgis@gmail.com (S.M.G.); 3Instituto de Investigación Biosanitaria de Granada (ibs.GRANADA), 18012 Granada, Spain; 4University of Navarre, Department of Preventive Medicine and Public Health, Medical School, 31008 Pamplona, Spain; ageas@unav.es (A.G.); mamartinez@unav.es (M.A.M.-G.); pilarbuilc@gmail.com (P.B.-C.); mcanela@unav.es (M.R.-C.); 5Navarra institute for health research (IdisNa), 31008 Pamplona, Spain; 6CIBER Fisiopatología de la Obesidad y Nutrición (CIBEROBN), Instituto de Salud Carlos III (ISCIII), 28029 Madrid, Spain; jordi.salas@urv.cat (J.S.-S.); dolores.corella@uv.es (D.C.); doraromaguera@yahoo.es (D.R.); luisfernando.aros@ehu.eus (F.A.); jwarnberg@uma.es (J.W.); jalfmtz@unav.es (J.A.M.); lluis.serra@ulpgc.es (L.S.-M.); restruch@clinic.ub.es (R.E.); fjtinahones@hotmail.com (F.J.T.); jlapetra@ono.com (J.L.); xpinto@bellvitgehospital.cat (X.P.); pep.tur@uib.es (J.A.T.); angarios2004@yahoo.es (A.G.-R.); pilar.matia@gmail.com (P.M.); clotilde.vazquez@fjd.es (C.V.); eros@clinic.ub.es (E.R.); andres.diaz@urv.cat (A.D.-L.); rebeca.fernandez@uv.es (R.F.-C.); mfito@imim.es (M.F.); jadzia.konieczna@gmail.com (J.K.); angelmago13@gmail.com (Á.M.A.-G.); iabetego@unav.es (I.A.); almudena.sanchez@ulpgc.es (A.S.-V.); rcasas1@clinic.cat (R.C.); araceli.munoz.garach.sspa@juntadeandalucia.es (A.M.-G.); josemanuel.santos@ono.com (J.M.S.-L.); jbasora.tarte.ics@gencat.cat (J.B.); Olga.portoles@uv.es (O.P.); mmonyino@gmail.com (M.M.); apalau@grupsagessa.com (A.P.G.); 7Department of Nutrition, Harvard T.H. Chan School of Public Health, Boston, MA 02115, USA; 8Universitat Rovira i Virgili, Departament de Bioquímica i Biotecnologia, Unitat de Nutrició, 43002 Reus, Spain; 9Institut d’Investigació Sanitària Pere Virgili (IISPV), 43204 Reus, Spain; 10University Hospital of Sant Joan de Reus, Nutrition Unit, 43204 Reus, Spain; 11Department of Preventive Medicine, University of Valencia, 46010 Valencia, Spain; 12Unit of Cardiovascular Risk and Nutrition, Institut Mar d´Investigacions Mèdiques (IMIM), 08003 Barcelona, Spain; mariadoloreszf@blanquerna.edu (M.D.Z.); kperez@imim.es (K.A.P.-V.); 13Human Nutrition Unit, Blanquerna-Ramon Llull University, 08001 Barcelona, Spain; 14Health Research Institute of the Balearic Islands (IdISBa), 07120 Palma de Mallorca, Spain; 15Nutritional Epidemiology Unit, Miguel Hernández University, ISABIAL-FISABIO, 03202 Alicante, Spain; 16Department of Cardiology, OSI ARABA, University Hospital Araba, University of the Basque Country UPV/EHU, 48940 Vitoria-Gasteiz, Spain; anai.m.rodriguez@gmail.com; 17Department of Nursing, School of Health Sciences. University of Malaga- Instituto de Investigación Biomédica de Málaga (IBIMA), 29010 Málaga, Spain; 18Department of Nutrition, Food Sciences, and Physiology, Center for Nutrition Research, University of Navarra, 31008 Pamplona, Spain; 19Nutritional Genomics and Epigenomics Group, IMDEA Food, CEI UAM + CSIC, 28049 Madrid, Spain; lidia.daimiel@imdea.org; 20Research Institute of Biomedical and Health Sciences. University of Las Palmas de Gran Canaria, 35016 Las Palmas de Gran Canaria, Spain; 21Department of Internal Medicine, Institut d´Investigacion Biomèdiques August Pi Sunyer (IDIBAPS), Hospital Clinic, University of Barcelona, 08036 Barcelona, Spain; 22Virgen de la Victoria Hospital, Department of Endocrinology, Instituto de Investigación Biomédica de Málaga (IBIMA), University of Málaga, 29016 Málaga, Spain; 23Department of Family Medicine, Research Unit, Distrito Sanitario Atención Primaria Sevilla, 41013 Sevilla, Spain; 24Lipids and Vascular Risk Unit, Internal Medicine, Hospital Universitario de Bellvitge, Hospitalet de Llobregat, 08907 Barcelona, Spain; 25Research Group on Community Nutrition & Oxidative Stress, University of Balearic Islands, 07122 Palma de Mallorca, Spain; laura.gallardo@uib.es; 26Lipids and Atherosclerosis Unit, Department of Internal Medicine, Maimonides Biomedical Research Institute of Córdoba (IMIBIC), Reina Sofía University Hospital, University of Córdoba, 14004 Córdoba, Spain; 27Department of Health Sciences, University of Jaen, 23071 Jaen, Spain; 28Department of Endocrinology and Nutrition, Instituto de Investigación Sanitaria Hospital Clínico San Carlos (IdISSC), 28040 Madrid, Spain; 29Institute of Biomedicine (IBIOMED), University of León, 24071 León, Spain; 30Department of Endocrinology, Institut d‘ Investigacions Biomédiques August Pi Sunyer (IDIBAPS), Hospital Clinic, University of Barcelona, 08036 Barcelona, Spain; jovidal@clinic.cat; 31CIBER Diabetes y enfermedades metabólicas (CIBERDEM), ISCIII, 28029 Madrid, Spain; 32Department of Endocrinology, Fundación Jiménez-Díaz, 28040 Madrid, Spain; 33Lipid Clinic, Department of Endocrinology and Nutrition, IDIBAPS, Hospital Clinic, 08036 Barcelona, Spain; 34Primary Care, Health Service of Navarra-Osasunbidea, 31002 Pamplona, Spain; 35Unidad Gestión Clínica de Prevención, Promoción y Vigilancia de la Salud, Distrito Atención Primaria Costa del Sol, Servicio Andaluz de Salud. Red de Investigación Servicios De Salud en Enfermedades Crónicas (REDISSEC), 29651 Málaga, Spain; eugenio.contreras.sspa@juntadeandalucia.es; 36Department of Medicine. University of Sevilla, 41004 Sevilla, Spain; 37Primary Care Division of Barcelona, Institut Català de la Salud-IDIAP Jordi Gol, 08007 Barcelona, Spain; mamunoz.bcn.ics@gencat.cat; 38ABS Reus V. Centre d’Assistència Primària Marià Fortuny, SAGESSA, 43203 Reus, Spain

**Keywords:** dietary diversity, nutrient adequacy, metabolic syndrome, aging, PREDIMED-Plus study

## Abstract

Dietary guidelines emphasize the importance of a varied diet to provide an adequate nutrient intake. However, an older age is often associated with consumption of monotonous diets that can be nutritionally inadequate, increasing the risk for the development or progression of diet-related chronic diseases, such as metabolic syndrome (MetS). To assess the association between dietary diversity (DD) and nutrient intake adequacy and to identify demographic variables associated with DD, we cross-sectionally analyzed baseline data from the PREDIMED-Plus trial: 6587 Spanish adults aged 55–75 years, with overweight/obesity who also had MetS. An energy-adjusted dietary diversity score (DDS) was calculated using a 143-item validated semi-quantitative food frequency questionnaire (FFQ). Nutrient inadequacy was defined as an intake below 2/3 of the dietary reference intake (DRI) forat least four of 17 nutrients proposed by the Institute of Medicine (IOM). Logistic regression models were used to evaluate the association between DDS and the risk of nutritionally inadequate intakes. In the higher DDS quartile there were more women and less current smokers. Compared with subjects in the highest DDS quartile, those in the lowest DDS quartile had a higher risk of inadequate nutrient intake: odds ratio (OR) = 28.56 (95% confidence interval (CI) 20.80–39.21). When we estimated food varietyfor each of the food groups, participants in the lowest quartile had a higher risk of inadequate nutrient intake for the groups of vegetables, OR = 14.03 (95% CI 10.55–18.65), fruits OR = 11.62 (95% CI 6.81–19.81), dairy products OR = 6.54 (95% CI 4.64–9.22) and protein foods OR = 6.60 (95% CI 1.96–22.24). As DDS decreased, the risk of inadequate nutrients intake rose. Given the impact of nutrient intake adequacy on the prevention of non-communicable diseases, health policies should focus on the promotion of a healthy varied diet, specifically promoting the intake of vegetables and fruit among population groups with lower DDS such as men, smokers or widow(er)s.

## 1. Introduction

Metabolic syndrome (MetS), a clustering of risk factors (central obesity, insulin resistance, dyslipidemia and hypertension) [1], is a well-known condition in the causal pathway of cardiovascular disease (CVD). MetS has also been associated with a higher risk of other chronic diseases, such as cancer [2] and neurodegenerative diseases [3]. In recent years, the prevalence of MetS has increased worldwide to the point that presently it is considered as a major public health problem [4]. This trend has also been observed in Spain, where the current prevalence of MetS is approximately 22.7% and increases with age [5]. Typically subjects with MetS have a higher use of health care services, incrementing costs [6].

MetS is a multifactorial disease that may be associated with some modifiable risk behaviors, such as unhealthy lifestyles and dietary patterns [7]. Among these factors, dietary intake plays a critical role in the prevention and treatment of MetS. Thus, dietary patterns that include healthy varied food groups and which provide adequate nutrient intake have been shown to be beneficial in the progression of MetS [8]. In this sense, the Mediterranean dietary pattern (MedDiet) has been related, not only to a delay in the progression and a lower mortality of MetS [9], but also to an adequate nutritional intake [10]. This can be explained by the great variety of food products that characterize the MedDiet. These foods, such as fruit and vegetables, nuts, legumes, fish and whole grain cereals, have a relatively low caloric value but a high nutrient content, increasing the probability to meet nutritional requirements [11].

Spanish dietary guidelines have emphasized the importance of a varied, balanced and moderated diet to reduce the risk of diet-related chronic diseases [12]. However, the role of a varied diet on chronic disease development is still uncertain. Some studies have suggested that dietary diversity (DD) contributes to high energy consumption and has a positive association with a poor quality diet, increasing the risk of MetS in older adults [13,14,15,16]; other researchers have reported that DD is a key component of high-quality diets, being associated with nutrient adequacy [17] and reducing the rates of CVD [18] and MetS [19] in the overall population.

Older adult populations with chronic diseases are considered vulnerable groups, as they are at greater nutritional risk due to a higher prevalence of inadequate nutrient intakes [20]. This could be a consequence of the consumption of monotonous and nutritionally inadequate diets, influenced by several factors, including loneliness, low socioeconomic status and functional quality [21].

There is evidence that nutritional inadequacy is prevalent in the older Spanish population [22], likely related to a monotonous diet and which could accelerate the progression of chronic diseases such as MetS. In this study we examined DD among PREDIMED-Plus participants, an older adult population with MetS, with the aim of assessing the association between DD and nutrient adequacy and to identify demographic variables associated with DD.

## 2. Materials and Methods

### 2.1. Design of the Study

A cross-sectional analysis on baseline data of the PREDIMED-Plus study was conducted. The PREDIMED-Plus study is an ongoing multicenter, randomized and parallel-group primary cardiovascular prevention trial. The PREDIMED-Plus study aims to assess the potential advantages of the synergy of a high-quality energy reduced MedDiet plus a weight-loss intervention and behavioral support on the incidence of CVD, in comparison to standard MedDiet advice (control group). The participant recruitment methods and data collection process have been described previously [23]. The Institutional Review Boards of all participating centers approved the study protocol. The clinical trial was registered in 2014 at the International Standard Randomized Controlled Trial (www.isrctn.com/ISRCTN89898870). All participants provided written informed consent.

### 2.2. Study Population

The study participants were men and women (55–75 years old and 60–75 years old, respectively), with overweight or obesity (body mass index (BMI) ≥27 and ≤40 kg/m^2^), who at baseline met at least three of the MetS criteria. The MetS criteria used have been previously described [24].

A total of 6874 subjects were recruited and randomized in 23 centers of the PREDIMED-Plus clinical trial from October 2013 to December 2016, from different universities, hospitals and research institutes across Spain. Of these, 287 participants were excluded for the present study (Figure 1): 47 participants because they did not complete the food frequency questionnaire (FFQ), and 240 participants because they reported values for total energy intake outside predefined limits (<3347 kJ <800 kcal/day or >17,573 kJ >4000 kcal/day for men); (<2510 kJ <500 kcal/day or >14,644 kJ >3500 kcal)/day for women) [25]. A final sample of 6587 participants was analyzed.

### 2.3. Dietary Intake Assessment

Trained dieticians collected data on dietary intake at baseline in a face-to-face interview. Dietary intake was assessed using a 143 item semi-quantitative FFQ previously and repeatedly validated in Spain [26]. The FFQ provides a list of foods commonly used by the Spanish population and asks about the consumption of these foods during the previous year. It includes nine response options (never or almost never, 1–3 times a month, once a week, 2–4 times a week, 5–6 times a week, once a day, 2–3 times a week day, 4–6 times a day and more than 6 times a day). The indicated frequencies of consumption were converted to intakes per day and multiplied by the weight of the standard serving size in order to estimate the intake in grams per day. Nutrient information was derived from Spanish food composition tables [27,28].

### 2.4. Dietary Diversity Score Construction

Using the 143-item validated FFQ mentioned above, we calculated an energy-adjusted DD score (DDS). This DDS was calculated by the method originally developed by Kant et al. [29] and recently used by Farhangi et al. [30]. We included five food groups: Vegetables, fruits, cereals, dairy products and protein food groups (legumes, meat, fish, eggs and nuts), based on the food groups recommended by the Spanish guidelines’ pyramid [12]. The vegetable group was divided into four subgroups, including: Green vegetables, tomatoes, yellow vegetables and mushrooms. The cereal group included potatoes and refined or whole grain cereals (bread, pasta, rice and breakfast cereals). The fruit group included all fresh fruit products divided in three categories: Citrus fruits, tropical fruits and other seasonal fruits. The dairy group included all kinds of milk, yogurt and cheese. Protein food groups included legumes (peas, beans, lentils and chickpeas), white meats (poultry and rabbit), fish (oily fish, white fish and other shellfish/seafood), eggs and nuts. Non-recommended food groups (that should be consumed as little as possible) [27], including sugar food groups (pastries, pies, biscuits, chocolate, fruit in sugar syrup and fruit juices) and food groups with high salt and/or saturated fats (butter, margarine, unhealthy vegetable fats, red meat, processed meats, sauces, pre-cooked dishes, condiments and snacks) were not included in the analysis as they are less healthy products and their variety is not desirable. These groups were used to define food variety groups. Therefore, we only analyzed diversity of recommended food groups, because the more important question was the percentage of total energy supplied by these food groups and our analyses were adjusted for total energy intake.

To be counted as a consumer for any of the food group categories reported previously, a subject should consume at least half of the recommended serving during one day (for example, if the Spanish nutritional recommendation advises a usual protein intake of three servings per week, for each protein item, participants should consume at least 1.5/7 servings/day). Within each food group, we summed up the number of items consumed. Each of the five predefined food categories received a maximum diversity score of 2 points, therefore the sum was rescaled to a 0-to-2 score by multiplying the score by 2 and dividing by the maximum score in that food group. Total DDS is the sum of the scores of the five main groups, theoretically ranging between 0 and 10 points. The score was adjusted for total energy intake, due to the general concern that high food variety might be a consequence of overconsumption of energy [14]. Finally, DDS was categorized in quartiles (Q) and the cutoff points were 3.8, 4.6 and 5.4. The variety in each food group was classified into four categories (C): C1 = 0 points, C2 ≥0–≤0.5 points, C3 ≥0.5–<1 points and C4 ≥1 point.

Subjects were asked about MedDiet adherence using a 17-item screening questionnaire used to both evaluate compliance with the intervention and guide the motivational interviews during the study follow-up. This screener is a modification of a previously validated 14-item MedDiet adherence questionnaire [31]. Compliance with each of the 17 items relating to characteristic food habits was scored with 1 point, and 0 points otherwise, so that the total score range was 0–17, with 0 meaning no adherence and 17 meaning maximum adherence. Adherence to the MedDiet, was categorized in tertiles as lower level (1st tertile, ≤7 points), medium (2nd tertile, 8–10 points) or higher level of adherence (3rd tertile, ≥11 points).

### 2.5. Nutrient Adequate Intake

The dietary intake of 17 selected nutrients, including vitamins A, B_1_, B_6_, B_9_, B_12_, C, D, E, minerals such as calcium, phosphorus, magnesium, iron, iodine, potassium, selenium and zinc and dietary fiber, was compared with age and sex-specific recommended intakes for these nutrients according to the established dietary reference intake (DRI) recommendations for the North-American population [32]. DRI is the general term for a set of reference values used to plan and assess nutrient intakes for healthy people. These values vary by age and sex. Intake levels above DRI imply a low likelihood of inadequate intake. To decrease potential measurement errors derived from the use of the FFQ, we calculated the proportion of individuals with intakes below two thirds (2/3) of the DRIs [33]. Furthermore, we estimated the proportion of inadequate intake according to European Food Safety Agency (EFSA) average requirements (ARs), taking as reference adequate intake (AI) when ARs were not available [34]. Results were based on dietary intake data only, excluding supplements.

### 2.6. Assessment of Non-Dietary Variables

At the baseline visit, trained PREDIMED-Plus staff collected information on lifestyle variables, educational achievement and socioeconomic status. The variables collected were sex, age (55–70 years and >70 years), educational level (primary, secondary and tertiary level, which includes university studies), civil status (married, widowed, divorced/single or other, which includes single participants and those who are priests or nuns who were categorized as “religious”) and whether they lived alone or not. Other lifestyle variables such as smoking habit (non-smoker, current smoker or never smoker), alcohol intake (measured as a continuous variable and expressed as intake in g/day) and physical activity (less active, moderately active and active) were taken into account. Individuals were classified based on their level of physical activity using a validated Spanish version of the Minnesota questionnaire: Less active (<4 MET), moderately active (4–5.5 MET) and active (≥6 MET) physical activity level [35,36]. Anthropometric variables (weight, height and waist circumference) were determined by trained staff in accordance with the PREDIMED-Plus operations protocol. Weight and height were measured with calibrated scales and a wall-mounted stadiometer, respectively. BMI was calculated as the weight in kilograms divided by the height in meters squared. Waist circumference (WC) was measured midway between the lowest rib and the iliac crest using an anthropometric tape.

### 2.7. Statistical Analysis

Data were analyzed using Stata (12.0, StataCorp LP, College Station, TX, USA). We used the PREDIMED-Plus baseline database generated in August 2017. Participants were classified according to DDS quartiles. Baseline characteristics of participants were described as means ± standard deviations (SD) for continuous variables or number and percentages for categorical variables. Comparison of quantitative variables across DDS quartiles was performed using ANOVA. Pearson χ^2^ test was used to compare the distribution of qualitative variables among DDS quartiles. A linear regression model was fitted to estimate the association of sociodemographic and lifestyle variables (sex, age, educational level, civil status, living alone, physical activity, smoking and drinking status) with DDS. Logistic regression models were used to evaluate the association between nutritional inadequate intakes (≥4 nutrients) as dependent variable and total DDS or food variety groups as the main independent variables. All analyses were adjusted for potential confounders based on prior knowledge: Sex, age, energy intake, BMI, WC, level of education, smoking status, physical activity, MedDiet adherence, marital status and living alone. We used a significance level of 0.05 for all analyses.

## 3. Results

### 3.1. Baseline Characteristics of PREDIMED-Plus Participants by Quartiles of DDS

Table 1 shows the comparison of demographic, anthropometric and lifestyle variables according to quartiles of DDS. We found significant differences for age, sex, smoking habits, marital status, educational level and WC (*p* < 0.001), but not for physical activity or BMI. Participants in the top DDS quartile (Q4) were older (65.8 ± 4.7) and more likely to be women (63.6%), never smokers (54.3%) and married (77.8%, *p* < 0.05) in comparison with the lower DDS quartiles. Moreover, participants in the top DDS quartile had a lower WC (106.3 ± 9.6) and lower educational level. However the magnitude of the differences across quartiles was small and should be interpreted in the light of the large power and sample size of the study.

### 3.2. Associations Between Demographic and Lifestyle Variables with DDS

The associations between these demographic and lifestyle variables with DDS as a continuous variable are presented in Table 2. We observed that DDS was significantly higher among women (mean difference = 0.26, 95% CI 0.18, 0.33), non-smokers (mean differences = 0.18, 95% CI 0.09, 0.27) and participants with higher adherence to the MedDiet (mean difference = 0.65, 95% CI 0.58, 0.73), whereas alcohol intake (mean difference = −0.01, 95% CI −0.01, −0.01) and being widowed (mean difference = −0.15, 95% CI −0.26, −0.05) were inversely associated with higher DDS.

### 3.3. Adherence to MedDiet and Dietary Intake of PREDIMED-Plus Participants by Quartiles of DDS Adjusted by Energy

Comparing across DDS quartiles, individuals in Q4 had significantly higher MedDiet adherence, higher intake of dietary fiber, carbohydrates, proteins and polyunsaturated fat, but lower saturated fat intake (Table 3). Vitamin and mineral intake increased progressively across DDS quartiles (*p* < 0.001). However the magnitude of these differences across quartiles is small and should be interpreted in the light of the large power and sample size of the study. On the other hand, participants in the bottom DDS quartile (Q1) reported higher alcohol intake. Total energy intake followed a U-shaped line, higher in Q1 and Q4 than in Q2–Q3. Appendix A) shows the proportion of participants with an intake below 2/3 of DRIs by DDS quartiles and is stratified by sex and group of age. The prevalence of inadequate intake of all nutrients decreased across DDS quartiles in all age and sex strata, except for vitamin D in older individuals. Vitamins B1, B12, C, phosphorus, iron, potassium, selenium and zinc presented high number of categories with zero cases, as nearly all the prevalent cases of deficient intake were at Q1 of DDS (Appendix A.). These results were similar when the EFSA recommendations were used instead (Appendix A.).

### 3.4. Distribution of Participants ny Number of Nutrients below Adequate Intake According to the DDS by Age and Sex

Table 4 shows the prevalence of four or more inadequacies in nutrient intake according to DDS quartiles, stratified by sex and age. Independently of age and sex, we observed that participants with the highest DDS (Q4) showed a lower number of nutrient inadequacies (*p* < 0.001). Also the prevalence of four or more inadequacies in nutrient intake decreased across DDS quartiles (*p* < 0.001) regardless of age or sex. When we used the EFSA dietary recommendations, we obtained similar results (Appendix A.).

### 3.5. Multivariable Logistic Regression Model for Inadequate Intake of Four or More out Eight Micronutrients According to Food Group’s Diversity Intake and Total DDS Quartiles

The risk of inadequate intake of four or more nutrients increased in the lower DDS quartiles, regardless of the model we chose to adjust by (Table 5). The adjusted odds ratio (OR) of inadequate intake was 28.56 (95% CI 20.80–39.21) for Q1 compared to Q4. We analyzed the prevalence of inadequate intake according to the category of DD for each one of the included food groups and found the same trend for all of them except for the cereal food group. The groups showing the strongest association were vegetables and fruit. These results were comparable if the EFSA criteria were used to define inadequate intake (Appendix A.). The adjustment by age as a quantitative variable did not change the results (data not shown).

## 4. Discussion

The present study, conducted among older individuals with MetS, showed that the greater the DDS the lower the risk of inadequate nutrient intake. Characteristics associated with a lower DDS are male sex, any marital status other than married, smoking habit and alcohol intake. Special attention should be paid to patients with these characteristics as they are likely to have a lower DDS.

It is known that demographic characteristics influence diet quality. The influence of age and sex on DDS could be attributable to multiple factors, including psychological and mental health issues, poorer nutritional knowledge, lack of cooking skills and increased loneliness [21,37]. Our results are in line with other studies regarding the impact of sex on the variety of food choices: Women consume more varied diets than men, presumably due to the traditional role as housewives and culinary knowledge [38]. Regarding age, despite other authors noting that dietary variety declined with age [39], we have not found the same trend, probably because the percentage of participants with >70 years old was small in our sample. Living alone has also been traditionally considered as a risk factor for poor dietary habits, mainly due to lower diversity of food intake [40,41]. We found lower DDS for widowed and divorced people, but not for people living alone.

In addition, some studies have highlighted that lower levels of education and economic status predict lower dietary variety [42,43]. Our results are not consistent on this point. According to socioeconomic level, the economic factors could explain low consumption of foods such as fish, fruits and vegetables, which require more frequent purchase and consumption and can also be more expensive. The discrepancy might be attributed to the fact that these studies have been carried out in non-European countries with heterogeneous socioeconomic levels. In our study, most participants had similar economic capacity (they were mostly retired) with the economic differences among them being small. Furthermore, the distribution of our population reflects the social and demographic characteristics of the Spanish population born in the 1940s–1960s. In that context, women had limited opportunity to pursue high levels of formal education. As the percentage of women is greater in the top quartile of DDS, this could be an attributable factor that explains that subjects with a higher DDS have a lower educational level.

In literature, smoking status and alcohol intake are the most important lifestyles variables related to food choices among older adults. In our study, non-smokers and drinkers of low quantities of alcohol showed higher DDS. Several studies have reported that smoking and drinking status are directly associated with less variety of food choices and poor nutrient intakes, consistent with our results [44,45]. However these findings have not been supported by a study carried out in middle-aged adults in Japan, probably because of socio-cultural differences and small sample size [46].

Dietary guidelines worldwide have promulgated the benefits of a variety of dietary intake, mainly because it is easier to provide the necessary amount of nutrients with a highly diverse diet. This could be especially important for obesity and chronic disease management [47,48]. In this regard, monotonous diets usually imply unhealthy eating habits, as well as the worsening in the progression of certain diseases, for example CVD [49]. In spite of this, some observational studies have related the diversity of food intake to higher rates of obesity and poor nutritional adequacy in adults [13,16]. However, in the case-control study of Karimbeiki et al. [16], cases were chosen from participants attending an obesity treatment group and dietary intake referred to the previous year, hence it is difficult to know whether it was cause or consequence. The study of Jayawardena et al. [13] estimate a DDS from a 24-h food record and not from a FFQ, and their results are not adjusted for total energy intake. Thus, their findings are not comparable to our results.

A recent non-systematic review concluded that “the scientific evidence to date does not support benefits of greater dietary diversity for optimal diet quality or healthy weight” [50] pointing out a need for standardized, reliable measures defining what diet diversity is. In the current study, high DDS level (Q4) was directly associated with adequate nutrient intake (≥4 nutrients out of 8) even after adjustment for confounders such as sociodemographic and lifestyle variables. This association corroborates findings reported by other authors [17,51,52], emphasizing the need to increase diet variety, specifically in older adult populations, in order to achieve adequate nutrient intakes in these vulnerable groups. A variety of recommended foods, such as vegetables, fruit, cereals and dairy products, decreases the risk of inadequate nutrient intake, mainly because these foods group are rich in vitamins and minerals and other healthy nutrients such as dietary fiber [53].

Based on results obtained from the adjusted binary logistic regression analyses, the higher DDS of the majority of the food groups analyzed was inversely associated with the risk of inadequate intake of nutrients (≥4 nutrients), except for cereals. Probably because the cereal group included not only whole grains, but also potatoes and refined grains. These findings are consistent with previous studies reporting a low probability of inadequate nutrient intakes in consumers of a high variety of foods groups, including vegetables and fruit [54], dairy products and protein-rich foods [51,55,56]. The notion is that for people who eat less variety of the healthy food groups, the intake of several nutrients might be endangered. For example, although vegetables provide a considerable amount of dietary fiber and water, green vegetables provide vitamins B_9_, while yellow ones are rich in vitamin A and carotenes. Another example of variety within the same group is the protein group, which includes eggs, white meat, legumes, fish and nuts. This group is an excellent source of high-quality protein, minerals and vitamins. In particular, white meat is high in B-vitamins, while oily fish is rich in polyunsaturated fat and eggs provide vitamins D and E and minerals such as zinc, iron and iodine [27,28].

Our study has some limitations. First, the study sample is not representative of the general population. Due to the trial inclusion and exclusion criteria, only older adults with MetS were included. Second, we did not have data about income status; however, we recorded the education level and employment status, which are both reasonable “proxy” indicators of socioeconomic status. Third, the cross-sectional design of the study does not allow for inferring causality. Irrespective of the direction of the associations, the variables included in the analysis have a high potential to improve the nutrient intake in older populations and allow the detection of groups of individuals more prone to nutrient deficiencies (those with lower DDS). Fourth, we used a FFQ to measure dietary intakes. Despite that the FFQ used has been validated in an adult Spanish population and has a good reproducibility and validity [26], it might not be the ideal tool to measure micronutrient intake [57]. For this reason, we considered that there was an inadequacy only when the intake did not reach 2/3 of the DRIs, correcting the possible bias introduced by the FFQ and assuming in any case that the inadequate micronutrient intake would be higher than the estimated figures. Fifth, it is important to consider that, besides pharmacological treatments, chronic diseases entail changes in dietary habits and nutrient metabolism, which have not been assessed. Last, we have estimated a DDS following the methodology of Kant et al. [29]. However, we excluded non-recommended foods such as sweets, snacks, juices and sweet beverages and processed foods because these products are high-energy density foods, rich in sodium, sugar and saturated fat, and also low-nutrient density foods, thus we considered that the intake of these food groups could not increase dietary diversity [58]. A culinary fat group was also not taken into account because Spanish individuals consume olive oil almost exclusively [59].

Some strengths of our study are its large sample size (*n* = 6587) and the considerable amount of baseline information collected in a large ongoing primary prevention trial, using a standardized protocol that reduces information bias regarding reported food intakes, sociodemographic characteristics and lifestyles. The individual analysis of each food group’s diversity can help to determine which aspects may maximize diet quality. The follow-up of this cohort will allow for the identification of any association between dietary diversity and clinical, metabolic and cardiovascular outcomes. Our results focus on the promotion of a healthy varied diet, specifically promoting the intake of a variety of vegetables and fruits among population groups with lower DDS such as men, smokers or widow(er)s people.

## 5. Conclusions

According to our findings, older Spanish adults with MetS had a high risk of inadequate nutrient intake. As DDS decreased, risk of inadequate nutrients intake increased. The impact of nutrient intake adequacy on the prevention of chronic non-communicable diseases, mainly among population groups with lower DDS such as men, older or widow people is very likely to play a crucial role from a public health perspective.

## Figures and Tables

**Figure 1 nutrients-11-00958-f001:**
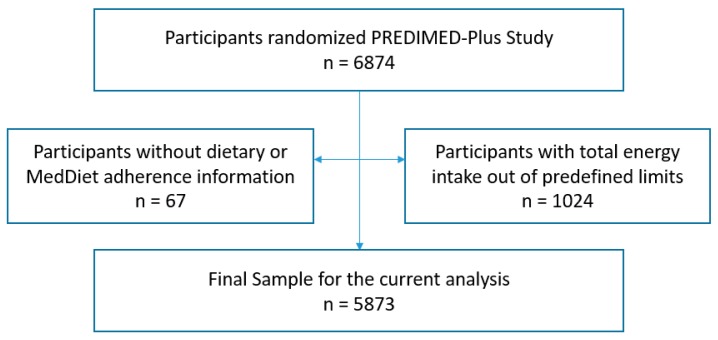
Flow-chart of participants.

**Table 1 nutrients-11-00958-t001:** Baseline characteristics of PREDIMED-Plus participants by quartiles of an energy-adjusted dietary diversity score (DDS, total population = 6587).

	Q1 (*n* = 1647)	Q2 (*n* = 1647)	Q3 (*n* = 1647)	Q4 (*n* = 1646)	*p* Value
**Age (Year), *n* (%)**					
55–70 years	1442 (87.6)	1416 (86.0)	1403 (85.2)	1356 (82.4)	<0.001
>70 years	205 (12.5)	231 (14.0)	244 (14.8)	290 (17.6)	
Mean ± SD	64.1 ± 5.1	64.8 ± 4.9	65.3 ± 4.8	65.8 ± 4.7	<0.001
**Sex, *n* (%)**					
Male	1116 (67.8)	916 (55.6)	770 (46.8)	600 (36.5)	<0.001
Female	531 (32.2)	731 (44.4)	877 (53.3)	1046 (63.6)	
**Smoking Habits, *n* (%)**					
Current Smoker	297 (18.0)	195 (11.8)	171 (10.4)	152 (9.2)	<0.001
Former Smoker	793 (48.2)	747 (45.4)	722 (43.8)	593 (36.0)	
Never Smoker	548 (33.3)	699 (42.4)	749 (45.5)	893 (54.3)	
Without information	9 (0.6)	6 (0.4)	5 (0.3)	8 (0.5)	
**Physical Activity, *n* (%)**					
Less active	1014 (61.7)	985 (59.9)	983 (59.9)	940 (57.4)	0.29
Moderately active	294 (17.9)	304 (18.5)	319 (19.4)	326 (19.9)	
Active	335 (20.4)	355 (21.6)	340 (20.7)	373 (22.8)	
**Educational Level, *n* (%)**					
Tertiary level	421 (25.6)	360 (21.9)	340 (20.6)	320 (19.5)	<0.001
Secondary level	534 (32.4)	480 (29.2)	449 (27.3)	435 (26.4)	
Primary level	679 (41.3)	796 (48.4)	842 (51.1)	873 (53.1)	
Without information	13 (0.7)	11 (0.6)	16 (1.0)	18 (1.0)	
**Civil Status, *n* (%)**					
Married	1258 (76.7)	1254 (76.4)	1243 (75.7)	1278 (77.8)	0.030
Widowed	151 (9.2)	162 (9.9)	182 (11.1)	186 (11.3)	
Divorced/Separated	145 (8.8)	123 (7.5)	130 (7.9)	117 (7.1)	
Others ^a^	93 (5.2)	108 (6.2)	92 (5.3)	65 (3.8)	
Living alone, n (%)	193 (11.8)	189 (11.5)	220 (13.4)	212 (12.9)	0.29
BMI (kg/m^2^), Mean ± SD	32.6 ± 3.4	32.5 ± 3.4	32.5 ± 3.5	32.5 ± 3.5	0.82
WC (cm), Mean ± SD	109.3 ± 9.5	108.1 ± 9.5	107.0 ± 9.7	106.3 ± 9.6	<0.001

Values are presented as means ± SD for continuous variables and *n* (%) for categorical variables. Pearson’s chi-square test was performed for categorical variables and ANOVA test for continuous variables. ^a^ includes religious and single status. Abbreviations: BMI, body mass index; DDS, dietary diversity score; Q, quartile; SD, standard deviation; WC, waist circumference.

**Table 2 nutrients-11-00958-t002:** Linear regression model to evaluate demographic and lifestyle variables associated with DDS (DDS measure as continuous variable).

	Total DDS	
	Mean Differences (95% CI)	*p* Value
**Sex**		
Men	0 (ref)	
Women	0.26 (0.18, 0.33)	<0.001
**Age**		
≤70 years	0 (ref)	
More 70 years	0.06 (−0.15, 0.14)	0.12
**Smoking Habits**		
Current smoker	0 (ref)	
Former smoker	0.14 (0.06, 0.23)	0.001
Never smoker	0.18 (0.09, 0.27)	<0.001
**Physical Activity Status**		
Less active	0 (ref)	
Moderate active	0.06 (−0.01, 0.13)	0.11
Active	0.03 (−0.04, 0.10)	0.34
**Educational Status**		
Tertiary Level	0 (ref)	
Secondary Level	0.02 (−0.05, 0.09)	0.59
Primary Le	0.07 (0.03, 0.14)	0.041
**MedDiet Adherence**		
Low Adherence	0 (ref)	
Medium Adherence	0.34 (0.28, 0.40)	<0.001
High Adherence	0.65 (0.58, 0.73)	<0.001
**Civil Status**		
Married	0 (ref)	
Widowed	−0.15 (−0.26, −0.05)	0.004
Divorced/Separated	−0.12 (−0.23, −0.02)	0.026
Others ^a^	−0.21 (−0.34, −0.08)	0.002
Living alone	−0.31 (−0.14, 0.07)	0.57
WC (cm) ^b^	−0.01 (−0.01, 0.01)	0.57
BMI (kg/m^2^) ^b^	−0.01 (−0.01, 0.01)	0.75
Alcohol intake (g) ^b^	−0.01 (−0.02, −0.01)	<0.001

Linear regression model (95% CI) for the DDS as a dependent variable according to baseline characteristics of participants. ^a^ Others: Includes religious and single status. ^b^ 1-unit increase. Abbreviations: BMI, body mass index; CI, confidence interval; DDS, dietary diversity score; SD, standard deviation; WC, waist circumference.

**Table 3 nutrients-11-00958-t003:** Adherence to MedDiet, mean energy, alcohol and nutrient intakes of PREDIMED-Plus participants by quartiles of DDS adjusted by energy.

	Q1 (*n* = 1647)	Q2 (*n* = 1647)	Q3 (*n* = 1647)	Q4 (*n* = 1646)	*p* Value
**MedDiet Adherence, *n* (%)**					
Low adherence	857 (52.0)	638 (38.7)	507 (30.8)	372 (22.6)	<0.001
Medium adherence	602 (36.7)	686 (41.7)	717 (43.5)	684 (41.6)	
High adherence	188 (11.4)	323 (19.6)	423 (25.7)	590 (35.8)	
Mean ± SD	7.4 ± 2.5	8.3 ± 2.6	8.8 ± 2.5	9.5 ± 2.6	<0.001
**Nutrient Intake, Mean ± SD**					
Total energy (Kcal/day)	2382.3 ± 612.8	2345.0 ± 557.9	2340.7 ± 527.0	2397.1 ± 502.6	0.006
Total fat intake (%)	39.9 ± 7.1	39.9 ± 6.6	39.5 ± 6.3	38.9 ± 5.9	<0.001
Monounsaturated fat (%)	20.7 ± 4.9	20.8 ± 4.7	20.5 ± 4.6	20.1 ± 4.3	0.002
Polyunsaturated fat (%)	6.2 ± 1.9	6.3 ± 1.9	6.4 ± 1.8	6.6 ± 1.8	<0.001
Saturated fat (%)	10.1 ± 2.2	10.1 ± 2.1	9.9 ± 1.9	9.7 ± 1.8	<0.001
Carbohydrate intake (%)	40.1 ± 7.5	40.2 ± 7.0	40.7 ± 6.5	41.3 ± 6.1	<0.001
Protein intake (%)	15.4 ± 2.6	16.6 ± 2.7	17.2 ± 2.8	17.9 ± 2.6	<0.001
Fiber intake (g/day)	21.4 ± 7.8	24.8 ± 7.7	27.1 ± 7.8	31.3 ± 8.8	<0.001
Vitamin A (µg/day)	909.3 ± 624.6	1075.1 ± 648.9	1133.2 ± 587.7	1302.9 ± 650.3	<0.001
Vitamin B1 (mg/day)	1.5 ± 0.4	1.6 ± 0.4	1.7 ± 0.4	1.8 ± 0.4	<0.001
Vitamin B6 (mg/day)	2.0 ± 0.5	2.2 ± 0.5	2.4 ± 0.5	2.7 ± 0.6	<0.001
Vitamin B9 (µg/day)	290.7 ± 82.3	335.6 ± 88.9	363.6 ± 90.9	416.1 ± 102.0	<0.001
Vitamin B12 (µg/day)	9.0 ± 4.5	9.7 ± 4.5	10.1 ± 4.3	10.9 ± 4.5	<0.001
Vitamin C (mg/day)	147.5 ± 66.4	189.0 ± 74.3	216.5 ± 78.3	255.7 ± 83.2	<0.001
Vitamin D (µg/day)	5.3 ± 3.2	5.9 ± 3.3	6.4 ± 3.4	7.1 ± 3.6	<0.001
Vitamin E (mg/day)	9.6 ± 4.1	10.4 ± 3.9	10.7 ± 3.7	11.7 ± 3.8	<0.001
Calcium (mg/day)	876.6 ± 325.3	987.2 ± 325.1	1071.8 ± 320.3	1201.5 ± 326.2	<0.001
Phosphorus (mg/day)	1556.4 ± 389.7	1699.4 ± 388.3	1796.7 ± 389.7	1985.6 ± 397.1	<0.001
Magnesium (mg/day)	371.0 ± 99.3	403.7 ± 98.4	428.2 ± 98.8	479.4 ± 108.5	<0.001
Iron (mg/day)	15.2 ± 3.9	16.0 ± 3.8	16.6 ± 3.7	18.1 ± 3.9	<0.001
Iodine (µg/day)	242.9 ± 163.6	274.7 ± 153.1	297.0 ± 155.3	327.2 ± 151.1	<0.001
Potassium (mg/day)	3767.9 ± 858.1	4262.1 ± 880.0	4619.6 ± 929.3	5256.5 ± 1053.4	<0.001
Selenium (µg/day)	113.6 ± 36.8	115.9 ± 32.9	116.4 ± 31.3	122.5 ± 30.9	<0.001
Zinc (mg/day)	12.5 ± 3.5	13.0 ± 3.2	13.2 ± 3.1	14.1 ± 3.1	<0.001
Alcohol intake (g/day), Mean ± SD	16.5 ± 19.4	11.7 ± 15.0	9.1 ± 12.4	6.8 ± 9.8	<0.001

Values are presented as means ± SD for continuous variables and *n* (%) for categorical variables. Pearson’s chi-square test was performed for categorical variables and ANOVA test for continuous variables. Abbreviations: DDS, dietary diversity score; Q, quartile; SD, standard deviation.

**Table 4 nutrients-11-00958-t004:** Number of inadequacies and distribution of participants with ≥4 nutrients below 2/3 of the dietary reference intake (DRI) according to DDS by age and sex.

	**MEN: ≤70 years**				
	**Q1 (*n* = 787)**	**Q2 (*n* = 763)**	**Q3 (*n* = 973)**	**Q4 (*n* = 489)**	***p* Value**
Inadequacies, mean ± SD	3.0 ± 1.1	2.3 ± 1.1	2.0 ± 0.1	1.7 ± 0.7	<0.001 ^1^
Participants, *n* (%)	468 (46.8)	156 (19.3)	78 (11.7)	12 (2.4)	<0.001 ^2^
	**WOMEN: ≤70 years**				
	**Q1 (*n* = 630)**	**Q2 (*n* = 610)**	**Q3 (*n* = 884)**	**Q4 (*n* = 529)**	***p* Value**
Inadequacies, mean (SD)	2.9 ± 1.0	2.5 ± 1.1	2.1 ± 1.0	1.7 ± 0.8	<0.001 ^1^
Participants, *n* (%)	169 (38.2)	145 (23.8)	81 (11.0)	23 (2.7)	<0.001 ^2^
	**MEN: >70 years**				
	**Q1 (*n* = 124)**	**Q2 (*n* = 136)**	**Q3 (*n* = 111)**	**Q1 (*n* = 48)**	***p* Value**
Inadequacies, mean (SD)	2.9 ± 1.1	2.6 ± 1.2	2.0 ± 1.0	1.6 ± 0.9	<0.001 ^1^
Participants, n (%)	51 (44.0)	37 (33.9)	11 (10.9)	5 (5.5)	<0.001 ^2^
	**WOMEN: >70 years**				
	**Q1 (*n* = 138)**	**Q2 (*n* = 137)**	**Q1 (*n* = 177)**	**Q1 (*n* = 103)**	***p* Value**
Inadequacies, mean (SD)	3.0 ± 1.1	2.5 ± 1.1	2.0 ± 0.9	1.7 ± 0.8	<0.001 ^1^
Participants, n (%)	38 (42.7)	30 (24.6)	14 (9.8)	6 (3.0)	<0.001 ^2^

^1^*p* value: Pearson’s Chi-Square test was used to estimate differences among prevalence of inadequate nutrient intake according to quartiles of DDS for sex strata. ^2^
*p* value: ANOVA test was performed to estimate differences among mean of inadequate nutrient intakes according to sex, for each DDS quartile. Abbreviations: DDS, dietary diversity score; DRI, dietary reference intake; Q, quartile; SD, standard deviation.

**Table 5 nutrients-11-00958-t005:** Multivariable logistic regression models for inadequate intake of four or more out eight micronutrients according to food group’s diversity intake and total DDS quartiles in the PREDIMED-Plus study participants. Odds ratios (95% Confidence intervals).

	**Q1** **(*n* = 1647)**	**Q2 ** **(*n* = 1647)**	**Q3** **(*n* = 1647)**	**Q4** **(*n* = 1646)**
**Total DDS**				
Model 1	27.42 (20.13−37.34)	10.00 (7.30−13.72)	4.37 (3.14−6.09)	1 (Ref.)
Model 2	28.56 (20.80−39.21)	9.97 (7.25−13.70)	4.33 (3.11−6.04)	1 (Ref.)
	**C1** **(*n* = 550)**	**C2** **(*n* = 1315)**	**C3** **(*n* = 2482)**	**C4** **(*n* = 2240)**
**Vegetable Group**				
Model 1	19.82 (15.19−25.85)	7.28 (5.85−9.10)	2.74 (2.22−3.38)	1 (Ref.)
Model 2	14.03 (10.55−18.65)	6.21 (4.92−7.83)	2.52 (2.02−3.14)	1 (Ref.)
	**C1** **(*n* = 845)**	**C2** **(*n* = 4497)**	**C3** **(*n* = 779)**	**C4** **(v466)**
**Fruit Group**				
Model 1	19.75 (11.87−32.86)	3.76 (2.30−6.15)	2.23 (1.29−3.84)	1 (Ref.)
Model 2	11.62 (6.81−19.81)	2.71 (1.62−4.53)	2.02 (1.15−3.57)	1 (Ref.)
	**C1** **(*n* = 350)**	**C2** **(*n* = 4767)**	**C3** **(*n* = 1390)**	**C4** **(*n* = 80)**
**Cereal Group**				
Model 1	1.33 (0.54−3.31)	1.13 (0.47−2.71)	0.90 (0.37−2.19)	1 (Ref.)
Model 2	0.83 (0.32−2.19)	0.84 (0.33−2.14)	0.71 (0.28−1.82)	1 (Ref.)
	**C1** **(*n* = 26)**	**C2** **(*n* = 1254)**	**C3** **(*n* = 2770)**	**C4** **(*n* = 2537)**
**Proteins group**				
Model 1	12.33 (4.10−37.19)	3.00 (2.48−3.62)	2.00 (1.69−2.37)	1 (Ref.)
Model 2	6.60 (1.96−22.24)	2.02 (1.64−2.48)	1.63 (1.36−1.96)	1 (Ref.)
	**C1** **(*n* = 686)**	**C2** **(*n* = 2447)**	**C3** **(*n* = 2600)**	**C4** **(*n* = 854)**
**Dairy Group**				
Model 1	9.51 (6.88−13.14)	3.35 (2.50−4.49)	1.52 (1.12−2.06)	1 (Ref.)
Model 2	6.54 (4.64−9.22)	2.40 (1.76−3.27)	1.24 (0.90−1.71)	1 (Ref.)

Values are presented as OR and 95% CI for inadequate intake of micronutrients as categorical variable according to total DDS and food’s group diversity. Model 1: Adjusted for energy intake. Model 2: Adjusted for energy intake, sex, age, smoking habits, physical activity, educational level, MedDiet adherence, BMI, alcohol intake, living alone and civil status. Abbreviations: BMI, body mass index; C, category; DDS, dietary diversity score; Q, quartile.

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
