# Peer review of "Dietary Diversity and Nutritional Adequacy among an Older Spanish Population with Metabolic Syndrome in the PREDIMED-Plus Study: A Cross-Sectional Analysis"

_nutrients, 2019, doi:10.3390/nu11050958_

Round 1

Reviewer 1 Report

In the manuscript a dietary diversity (DD) and nutritional adequacy among older Spanish population with metabolic syndrome were studied. The manuscript is overload by tables and presented results, and some methodological aspects require clarification. Also the aim of the manuscript presented in the Abstract (page 2, line 86-88) and in the Introduction section (page 3, line 138-140) needs revision. Moreover, associations between DD and demographic variables should be determine using some more advanced analysis (for example: logistic regression analysis).

Detailed comments:

1.       Abstract (page 2, line 89): Was overweight/obesity included into MetS? The part of sentence ‘…. With overweight/obesity and MetS’ seems a bit confusing.

2.       Methods (page 5, line 198-207): the authors tried describe how calculated DDS. Although, they did it using as an example protein food groups, all time it is not sufficient clear. The authors did not explain how they treated other 4 food groups included to the DDS.

3.       Methods (page 5, line 210-217): Why the authors to examine an adherence to MedDiet? In the manuscript' title, aim and hypothesis it has not been included .

4.       Methods (page 6, line 256): I understand that age was included as a categorical variable in the models (55-70 y, and >70 y). I recommend to include it as continuous variable. It is a big difference in nutritional behaviors, health status etc. between 55 years old and 70 years old person – now these people are in the same group.

5.       Table 2: please check the correctness of iodine (it is typo) and units of alcohol intake.

6.       Table 3 and Table 4: In my opinion these tables are a bit obvious and predictable. Results presented in Table 3 may be treat as a sensitivity analysis and may be present only as an additional supplementary material. Results presented in Table 4 (without stratification by age) may constitute a summary of results presented in Table 2. Instead of these tables some global analysis, which allow to indicate associations between DD and demographic variables, should be use (for example: logistic regression analysis).

7.       Table 5:  Total DDS should be presented firstly before presenting specific food groups.

8.       Table 5 and Methods (page 5, line 207-209): it is not clear why categories were created and for example what does it mean C4 for vegetable group and C4 for cereal group? whether C4 for these groups reflect other or quite similar diversity of consumption.

9.       Discussion (page11, line 334): the authors concluded that widowed, smokers etc. may have at higher risk of DDS. In my opinion without comprehensive analysis,  it is not possible to make these conclusions.

My general advice is:

Revision of the aim of the study or conducting global analysis between DD and demographic variables should be include.

Author Response

RESPONSE LETTER: REVIEWER 1

1.    Abstract (page 2, line 89): Was overweight/obesity included into MetS? The part of sentence ‘…. With overweight/obesity and MetS’ seems a bit confusing.

Thank you for the time that you have spent reading and reviewing this article, as well as the positive and helpful comments. Our manuscript deals with the diet diversity and nutrient adequacy of subjects with Metabolic Syndrome. The inclusion criteria proposed in the PREDIMED-Plus study protocol (Martinez-Gonzalez M, Buil-Cosiales P, Corella D, Bulló M, Fito M, Vioque J, et al. Cohort Profile: Design and methods of the PREDIMED-Plus. International Journal of Epidemiology. 2018) specify that the recruited population study had to be subjects with overweight or obesity (BMI ranging from 27 to 40 kg/m2) and who also met at least three criteria for Metabolic Syndrome: raised blood pressure (systolic≥130 and/or diastolic ≥85 mm Hg), dyslipidaemia, that includes elevated triglycerides (≥150 mg/dL), reduced high-density lipoprotein cholesterol (<40 mg/dL in males; <50 mg/dL in females), raised fasting glucose (≥100 mg/dL), and/or central obesity (≥102 cm in men and ≥88 cm in women). These criteria have been previously described in literature (Alberti KGMM, Eckel RH, Grundy SM, Zimmet PZ, Cleeman JI, Donato KA, et al. Harmonizing the metabolic syndrome: A joint interim statement of the international diabetes federation task force on epidemiology and prevention; National heart, lung, and blood institute; American heart association; World heart federation; International atherosclerosis society; And international association for the study of obesity. Circulation. 2009; 120 (16):1640-5). This information was described in the Methodology (subsection: Study Population), (lines 159-160). Furthermore, we have included the word “also” in abstract (line 93) with the aim of facilitating the interpretation for the reader.

2.      Methods (page 5, line 198-207): the authors tried describe how calculated DDS. Although, they did it using as an example protein food groups, all time it is not sufficient clear. The authors did not explain how they treated other 4 food groups included to the DDS.

According to this comment, we tried to further explain how we calculated the Dietary Diversity Score. We explained that the food items in the Food Frequency Questionnaire were grouped into five major food groups: vegetables, fruits, cereals, proteins and dairy products. The classification was consistent with food grouping according to the Spanish Food Guide Pyramid (Aranceta Bartrina J, Arija Val VV, Maíz Aldalur E, Martínez de Victoria Muñoz E, Ortega Anta RM, Pérez-Rodrigo C, et al. Dietary Guidelines for the Spanish population (SENC, diciembre 2016); the new graphic icon of healthy food. Nutrición hospitalaria. 2016;33:1-48). To measure dietary variety, namely a Dietary Diversity Score (DDS), we developed a score named DDS. The DDS was developed based on the method originally developed by Kant et al. (Kant AK, Thompson FE. Measures of overall diet quality from a food frequency questionnaire: National Health Interview Survey, 1992. Nutrition Research. 1997; 17(9):1443-56). A variation of the above score was to examine the extent of consumption of foods recommended in current dietary guidance avoiding non-recommended food groups. To be counted as a “consumer” for any of the food group categories, a respondent needed to consume at least one-half serving as defined by the Spanish Food Pyramid quantity criteria for one day. Each of the five broad food categories received a maximum diversity score of 2 out of the 10 possible score points. The maximum and minimum scores of diversity were ranged between 0 and 10. Total score was the sum of the scores of the five main groups.  In order to facilitate the reader’s understanding, we wrote an example, taking as reference the protein food group. The remaining four food groups included were treated similarly.

3.      Methods (page 5, line 210-217): Why the authors to examine an adherence to MedDiet? In the manuscript' title, aim and hypothesis it has not been included.

Evaluating how MedDiet adherence affects the Dietary Diversity or Nutrient adequacy was not part of the study objectives. Therefore, our aim was to analyze how the Dietary Diversity could affect  the nutrient adequacy of older adults with Metabolic Syndrome. As we described in the introduction, dietary patterns like MedDiet, that include healthy varied food groups could provide adequate nutrient intake, due to the great variety of food products that characterize it. Because MedDiet adherence could be a confounding factor, we have taken it into account in adjusted regression models performed in our statistical analysis.

4.       Methods (page 6, line 256): I understand that age was included as a categorical variable in the models (55-70 y, and >70 y). I recommend to include it as continuous variable. It is a big difference in nutritional behaviors, health status etc. between 55 years old and 70 years old person – now these people are in the same group.

We are agree with this critique. However, we stratified our population according to sex and age in order to analyze the nutrient deficient intake, because nutritional requirements differ according to age (51-70 years, >70 years) and gender (Health NIo. Nutrient Recommendations: Dietary Reference Intakes (DRI) 2018. Available online: https://ods.od.nih.gov/Health_Information/Dietary_Reference_Intakes.aspx (accessed on 10 May 2018)). Nevertheless, we agree that although minimally, the inclusion of age as categorical variable in the regression models could change the results. Therefore, we have reproduced the analysis including age in the model as a continuous variable. As you can see in the table below, we did not find any significant differences regarding age compared to the previous analysis, and therefore the association between DDS and nutrient adequacy remained the same. Based on this, we decided to maintain the previous analysis. We have included in the main text (line 358-359) the following additional information: “The adjustment by age as a quantitative variable did not change the results (data not shown)”. Nevertheless, we could replace the table with the new analysis if you consider it as necessary.

Table 5. Multivariable logistic regression models for inadequate intake of 4 or more out 8 micronutrients according to food group’s diversity intake and total DDS quartiles in the PREDIMED-Plus study participants

C1

(n=550)

C2

(n=1315)

C3

(n=2482)

C4

(n=2240)

Vegetable   group

Model 1

17.99 (14.17-22.83)

7.34 (6.00-9.01)

3.10  (2.54-3.77)

1 (Ref.)

Model 2

19.82 (15.19-25.85)

7.28 (5.85-9.10)

2.74 (2.22-3.38)

1 (Ref.)

Model 3

15.42 (11.48-20.70)

6.55 (5.15-8.34)

2.64 (2.10-3.31)

1 (Ref.)

C1

(n=845)

C2

(n=4497)

C3

(n=779)

C4

(n=466)

Fruit   group

Model 1

25.30 (15.50-41.31)

5.41 (3.36-8.72)

2.73 (1.61-4.62)

1 (Ref.)

Model 2

19.75 (11.87-32.86)

3.76 (2.30-6.15)

2.23 (1.29-3.84)

1 (Ref.)

Model 3

1070 (6.25-18.33)

2.57 (1.53-4.32)

1.88 (1.06-3.35)

1 (Ref.)

C1

(n=350)

C2

(n=4767)

C3

(n=1390)

C4

(n=80)

Cereal  group

Model 1

8.83 (3.74-20.83)

3.32 (1.44-7.65)

1.62 (0.69-3.77)

1 (Ref.)

Model 2

1.33 (0.54-3.31)

1.13 (0.47-2.71)

0.90 (0.37-2.19)

1 (Ref.)

Model 3

0.76 (0.29-2.01)

0.85 (0.33-2.16)

0.72 (0.28-1.86)

1 (Ref.)

C1

(n=26)

C2

(n=1254)

C3

(n=2770)

C4

(n=2537)

Proteins  group

Model 1

31.76 (12.63-79.84)

5.24 (4.39-6.25)

2.74 (2.33-3.21)

1 (Ref.)

Model 2

12.33 (4.10-37.19)

3.00 (2.48-3.62)

2.00 (1.69-2.37)

1 (Ref.)

Model 3

6.27 (1.83-21.46)

2.13 (1.72-2.63)

1.68 (1.40-2.03)

1 (Ref.)

C1

(n=686)

C2

(n=2447)

C3

(n=2600)

C4

(n=854)

Dairy  group

Model 1

13.57 (10.00-18.42)

4.58 (3.45-6.01)

1.88 (1.40-2.51)

1 (Ref.)

Model 2

9.51 (6.88-13.14)

3.35 (2.50-4.49)

1.52 (1.12-2.06)

1 (Ref.)

Model 3

6.43 (4.52-9.16)

2.42 (1.76-3.33)

1.26 (0.91-1.76)

1 (Ref.)

Q1

 (n=1647)

Q2

(n=1647)

Q3

(n=1647)

Q4

(n=1646)

Total DDS

Model 2

27.42 (20.13-37.34)

10.00 (7.30-13.72)

4.37 (3.14-6.09)

1 (Ref.)

Model 3

29.07 (21.02-40.20)

9.60 (6.93-13.31)

4.26 (3.03-5.98)

1 (Ref.)

Model 1: Unadjusted. Model 2: Adjusted for energy intake. Model 3: Adjusted for energy intake, sex, age (continuous variable), smoking habits, physical activity, educational level, MedDiet adherence, BMI, alcohol intake, living alone and civil status.

5.      Table 2: please check the correctness of iodine (it is typo) and units of alcohol intake.

We have removed the term "iodo", replacing it with "iodine" in table 2. Moreover, we have also included in table 2 the units of alcohol intake (g/d), measured as continuous variable.

6.      Table 3 and Table 4: In my opinion these tables are a bit obvious and predictable. Results presented in Table 3 may be treat as a sensitivity analysis and may be present only as an additional supplementary material. Results presented in Table 4 (without stratification by age) may constitute a summary of results presented in Table 2. Instead of these tables some global analysis, which allow to indicate associations between DD and demographic variables, should be use (for example: logistic regression analysis).

Thank you for this critique. As our objective was to analyse how the DDS could affect the nutrient adequacy, we considered that it would be important to show how the prevalence of inadequate intake of the nutrients assessed decreased across DDS (table 3 of the manuscript) according to the nutritional requirements proposed by the Institute of Medicine (Health NIo. Nutrient Recommendations: Dietary Reference Intakes (DRI) 2018. Available online: https://ods.od.nih.gov/Health_Information/Dietary_Reference_Intakes.aspx (accessed on 10 May 2018)) and the European Food Safety Agency (EFSA. Dietary Reference Values and Dietary Guidelines  Available online: https://www.efsa.europa.eu/en/topics/topic/dietary-reference-values. (accessed on 10 May 2018). According to table 3, we have only included some nutrients intake according DDS quartile and other nutrients are provided as supplementary material (Supplementary table 2). However, as per your suggestion we have moved table 3 to supplementary material (now supplementary table 1).

Regarding the comment about tables 2 and 4,  we strongly believe that table 2 gives information about mean nutrients intake for the study population, whereas the table 4 contributes with information about the percentage of people with intake below the nutritional requirements. Moreover, we think that the information showed in table 4 is relevant in order to facilitate the interpretation of table 5. Therefore we have not made any modification.

Finally, we also recognize that demographic and lifestyle variables could exert an important effect on DDS, for this reason in our manuscript we performed a linear regression model in order to estimate the association of sociodemographic and lifestyle variables (sex, age, educational level, civil status, living alone, physical activity, smoking and drinking status) with total DDS as a continuous variable (previously included as Supplementary table 1). We have included this table in the main text (now Table 2).

7.      Table 5:  Total DDS should be presented firstly before presenting specific food groups.

We have restructured table 5. As the total DDS is made up of each of the specific five food group`s included in our analysis, we have now presented the total DDS first, trying to highlight the most interesting results.

8.      Table 5 and Methods (page 5, line 207-209): it is not clear why categories were created and for example what does it mean C4 for vegetable group and C4 for cereal group? Whether C4 for these groups reflect other or quite similar diversity of consumption.

The classification in categories used in the article were predefined a priori. Because the diversity of each of the specific five food group`s has not been used previously, there is no current agreement over the cut-off point that indicates it. The introduction of the index score as a continuous variable could provide a significant result, however, by dividing it into categories we obtain more intuitive results that are easy to explain. In the methods, we have now clarified the cut-off points for the categories with the aim of providing more information (line 218-220).

9.      Discussion (page11, line 334): the authors concluded that widowed, smokers etc. may have at higher risk of DDS. In my opinion without comprehensive analysis, it is not possible to make these conclusions.

We are aware of the necessity to establish associations between demographic and lifestyle variables and DDS (associations currently showed in Table 2). This clue can allow to select the most vulnerable people in order to prioritizing nutritional interventions. We rephrased the sentence to comply with your comments, that now makes more sense in the light of results presented in Table 2. You can read the new information at lines (370-375).

Reviewer 2 Report

The Authors aimed to assess the relationship between dietary diversity (DD) and nutrient adequacy and to identify associated demographic variables related with DD. They analysed cross-sectional baseline data from a previous study (PREDIMED-Plus study) among Spanish adults, aged 55–75 years, with overweight/obesity and MetS. Using a 143-item validated semi-quantitative food frequency questionnaire (FFQ). They established an energy-adjusted score of 90 DD (DDS), while defined as nutrient inadequacy an intake below 2/3 of the recommended dietary intake at least of ≥4 of 17 nutrients. By logistic regression models, they found that in the higher DDS quartile there were more women and less current smokers. Compared with subjects in the highest DDS quartile, those in the lowest DDS quartile had a higher risk of inadequate nutrient intake [OR=28.56 (95% C.I. 20.80-39.21)]. When estimated diversity for each one of the food groups, participants in the first quartile of diversity had a higher risk of nutrient deficiency: for vegetables, OR= 14.03 (IC 95% 10.55-18.65),  fruits OR=11.62 (IC 95% 6.81-19.81), dairy products OR= 6.54 (IC 95% 4.64-9.22) and protein foods OR=6.60 (IC 95% 1.96-22.24). As DDS decreases, the risk of inadequate nutrients intake rises. They concluded suggesting that health policies should focus on the promotion of a healthy varied diet, specifically promoting the intake of vegetables and fruit among population groups with lower DDS such as men, smokers or widow people.

The research topic is of interest. The study is well designed and written in an accetable way. Statistical analysis is appropriate. References are updated. The Authors recognize strenghts and limitations of the study. They conclude in an honest way. The paper is suitable for publication in the modest opinion of this reviewer in the present form.

Author Response

RESPONSE LETTER: REVIEWER 2

The Authors aimed to assess the relationship between dietary diversity (DD) and nutrient adequacy and to identify associated demographic variables related with DD. They analysed cross-sectional baseline data from a previous study (PREDIMED-Plus study) among Spanish adults, aged 55–75 years, with overweight/obesity and MetS. Using a 143-item validated semi-quantitative food frequency questionnaire (FFQ). They established an energy-adjusted score of 90 DD (DDS), while defined as nutrient inadequacy an intake below 2/3 of the recommended dietary intake at least of ≥4 of 17 nutrients. By logistic regression models, they found that in the higher DDS quartile there were more women and less current smokers. Compared with subjects in the highest DDS quartile, those in the lowest DDS quartile had a higher risk of inadequate nutrient intake [OR=28.56 (95% C.I. 20.80-39.21)]. When estimated diversity for each one of the food groups, participants in the first quartile of diversity had a higher risk of nutrient deficiency: for vegetables, OR= 14.03 (IC 95% 10.55-18.65),  fruits OR=11.62 (IC 95% 6.81-19.81), dairy products OR= 6.54 (IC 95% 4.64-9.22) and protein foods OR=6.60 (IC 95% 1.96-22.24). As DDS decreases, the risk of inadequate nutrients intake rises. They concluded suggesting that health policies should focus on the promotion of a healthy varied diet, specifically promoting the intake of vegetables and fruit among population groups with lower DDS such as men, smokers or widow people.

The research topic is of interest. The study is well designed and written in an accetable way. Statistical analysis is appropriate. References are updated. The Authors recognize strenghts and limitations of the study. They conclude in an honest way. The paper is suitable for publication in the modest opinion of this reviewer in the present form.

Thank you for your positive comments and critique of our manuscript and the time taken to review our manuscript. In order to improve our manuscript even more we have made some further changes and sent the manuscript to a native English speaker for review to improve its readability for the journal’s readers.

Reviewer 3 Report

Thank you for the opportunity to read and review the manuscript titled, Dietary diversity and nutritional adequacy among an older Spanish population with metabolic syndrome in the PREDIMED-Plus Study: a cross-sectional analysis. In this manuscript, the numerous authors reported on dietary findings, based on FFQ, in 6587 Spanish adults aged 55-75 years, who were overweight/obese and had metabolic syndrome (MetS). Although the overarching study seems to be reasonably well characterised, with a large study population, this particular manuscript contains numerous stylistic and scientific errors that must be attended to prior to moving forward.

The manuscirpt contains several syntax, grammatical and linguistic errors that, at times, are distraction, whereas at other times, impede comprehension of the text. I have not provided specific comments on all of the errors. Instead, the authors may wish for a native English speake to review and revise the manuscript.

The manuscript alternates between British English (e.g. analysed) and American English (emphasized). This is rather distracting, and reflects poor editing.

Respectfully, I would suggest that 55 years old is not aged! Find a better descriptor. 

Line 91: The authors use the abbreviation DDS without defining it.

Line 91: "...below 2/3 of the recommended dietary intake..." per which recommendations?

Line 92: This is a cross-sectional study, that uses logistic regression. As such, it is incorrect to describe "relationships." Rather, the authors are describing "assocations."

Line 99: The authors report an IC95%. In the results, this is described as a CI95%. However, it is much more common to use the abbreviation 95%CI. (At the very least, the abbreviation should be consistent throughout.)

Lines 154-161: This sentence is 7 lines long. It is far too long and complex. Even as a native English speaker, I got lost in it.

Lines 161-2: The authors write out cardiovascular disease. Yet, in Lines 131, and 147, they refer to CVD (undefined abbreviation, although understood). This is rather distracting, and reflects poor editing.

Line 196: If the authors are calculating diet (not dietary) diversity scores, I wonder why "non-recommended food groups (Lines 192-3) and food groups with high salt and/or saturated fats (Lines 195-6; note grammatical correction) were excluded. These food groups could have a negative contribution to the overall score.

Line 236: The authors provide three categories of physical activity (less active, moderately active, or active); and alcohol intake. Yet, neither the variables or the categorisation were defined. A similar comment applies to other demographic variables. However, these are more easily understood and thus do not necessarily need a definition (e.g. civil status, smoker).

Line 266: The authors write that participants in the top DDS quartile had lower WC (not surprising), and lower education (surprising!). A comment on the latter would be warranted in the discussion section.

Table 1: Whereas nearly all of the analyses presented are statistically significant, one does wonder about the meaningfulness about some of the differences. For example, the difference in mean age between the quartiles is sigificant (p<0.001), but corresponds to a mean difference of approximately 1.5 years. Considering the age range of the population is 55-75 years, what does this really mean? Is this difference relevant?

Line 277: What does "religious and single status" refer to? This was not described in the methods.

Table 2: Similar to the statistical differences reported in Table 1, one wonders about the meaningfulness of the statistical differences in some of the nutrients upon stratification by quartile.

Table 3: Interpretation of this table would have been easier if the authors had provided a n for each gender-age group.

Table 4: Please consider the presentation of this table. At present, the lengthy variable names are repeated several times. The table could be substantially edited for clarity of presentation. 

Table 5: Please provide labels for what are presumably OR and 95%CI. 

Table 5: Some of the point estimates change considerably with partial- and full adjustments. Consider if this is the the result of overestimation. If so, how could this be addressed?

Table 5: For Total DDS, why was no Model 1 presented?

Lines 332-334: In the summary sentence of the discussion, the authors note that "special attention" should be paid to several groups with lower DDS. Yet, it is not clear to which quartiles the authors are referring. For example, amongst the widow/widowers, the biggest difference is seen between Q2 and Q3. For other groups, such as alcohol drinkers, interpretation is hindered by the lack of definition of variables and the categorisation of the variables. 

Lines 365-369: If these studies cannot be compared to your study, why mention them in detail, or at all?

Line 415: Do the authors mean N, rather than n? 

Author Response

RESPONSE LETTER: REVIEWER 3

 We appreciate the suggestions made by the reviewer regarding the stylistic and scientific errors. After carefully reading each of the parts of the article, we sent it to a native English speaker in order to review the language and grammar used in the article with the aim of making it more clear and intuitive for the journal´s readers.

1.      Respectfully, I would suggest that 55 years old is not aged! Find a better descriptor.

We agree with this comment. We have removed the term "aged", replacing it with "older adults" through the text. We have replaced it and we have activated the track changes with the new term in the new version of the manuscript.

2.      Line 91: The authors use the abbreviation DDS without defining it.

A brief description of the DDS was used in the abstract before we used the abbreviation. Specifically, we wrote: “Using a 143-item validated semi-quantitative food frequency questionnaire (FFQ), we calculated an energy-adjusted score of DD (DDS)” (line 94-95). With the aim of facilitating interpretation to the reader, we have rewritten the sentence. Currently appears as: “Using a 143-item validated semi-quantitative food frequency questionnaire (FFQ), we calculated an energy-adjusted Dietary Diversity Score (DDS)” (line 93-94). This information also appears at lines 186-187 of the manuscript.

3.      Line 91: "...below 2/3 of the recommended dietary intake..." per which recommendations?

As we described in methods section, we used the age and sex-specific recommended intakes for the nutrients included in our analysis from the recommendations of the Dietary Reference Intake (DRI] for the North-American population (Health NIo. Nutrient Recommendations: Dietary Reference Intakes (DRI) 2018. Available online: https://ods.od.nih.gov/Health_Information/Dietary_Reference_Intakes.aspx (accessed on 10 May 2018)) and the European Food Safety Agency (EFSA. Dietary Reference Values and Dietary Guidelines  Available online: https://www.efsa.europa.eu/en/topics/topic/dietary-reference-values. (accessed on 10 May 2018). The DRIs recommendations are now specified in the abstract (line 97).  DRI is the general term for a set of reference values used to plan and assess nutrient intakes for healthy people. These values vary by age and sex. Intake levels above DRI imply a low likelihood of inadequate intake. As we note in the limitations of our study, despite the FFQ used having been validated in an adult Spanish population and has a good reproducibility and validity (Martin-moreno JM, Boyle P, Gorgojo L, Maisonneuve P, Fernandez-rodriguez JC, Salvini S, et al. Development and validation of a food frequency questionnaire in Spain. International Journal of Epidemiology. 1993; 22(3):512-9) it might not be the ideal tool to measure micronutrient intake (Ortiz-Andrellucchi A, Sánchez-Villegas A, Doreste-Alonso J, de Vries J, de Groot L, Serra-Majem L. Dietary assessment methods for micronutrient intake in elderly people: a systematic review. The British journal of nutrition. 2009;102 Suppl 1:S118-49).  For this reason, we considered that there was an inadequacy only when the intake did not reach 2/3 of the DRIs, correcting the possible bias introduced by the FFQ and assuming in any case that the inadequate micronutrient intake should be higher than the estimated figures. This criteria have been proposed previously by other authors (Aranceta J, Serra-Majem L, Pérez-Rodrigo C, Llopis J, Mataix J, Ribas L, et al. Vitamins in Spanish food patterns: The eVe study. Public Health Nutrition. 2001;4(6A):1317-23) and also is the same as the criteria we used in a recent paper published in Nutrients (Cano-Ibáñez N, Bueno-Cavanillas A, Martínez-González MA, Corella D, Salas-Salvadó J, Zomeño MD, et al. Dietary intake in population with metabolic syndrome: Is the prevalence of inadequate intake influenced by geographical area? Cross-sectional analysis from PREDIMED-plus study. Nutrients. 2018;10(11)). In addition, we estimated the proportion of inadequate intake according to European Food Safety Agency (EFSA) average requirements (ARs), taking as reference adequate intake (AI) when ARs were not available.   

4.      Line 92: This is a cross-sectional study that uses logistic regression. As such, it is incorrect to describe "relationships." Rather, the authors are describing "associations."

We agree with this comment. We have modified this term through the manuscript`s text, rewriting the aim of the study in the abstract (line 90, line 98), introduction (line 146-147), methodology (line 272) and discussion (line 473).

5.      Line 99: The authors report an IC95%. In the results, this is described as a CI95%. However, it is much more common to use the abbreviation 95%CI. (At the very least, the abbreviation should be consistent throughout.)

We have corrected the abbreviated term throughout and now appears as 95%CI.

6.      Lines 154-161: This sentence is 7 lines long. It is far too long and complex. Even as a native English speaker, I got lost in it.

We are agree with this suggestion. The sentence was too long and complex. We have restructured the sentence as follows: “The study participants were men and women (55-75 and 60-75 years old, respectively) with overweight or obesity (body mass index (BMI) ≥27 and ≤40 kg/m2) who at baseline met at least three components of the MetS. The MetS criteria used have been previously described [24]”. This information has been added in the main text (lines 161-163).

7.      Lines 161-2: The authors write out cardiovascular disease. Yet, in Lines 131, and 147, they refer to CVD (undefined abbreviation, although understood). This is rather distracting, and reflects poor editing.

We have carefully checked the abbreviations through the main text. In this sense, we have ensured that each term has been described before its inclusion as an abbreviation.

8.      Line 196: If the authors are calculating diet (not dietary) diversity scores, I wonder why "non-recommended food groups (Lines 192-3) and food groups with high salt and/or saturated fats (Lines 195-6; note grammatical correction) were excluded. These food groups could have a negative contribution to the overall score.

We really appreciate your suggestions. We have written the manuscript according to the references that we take as a model to estimate DDS (Kant AK, Schatzkin A, Harris TB, Ziegler RG, Block G. Dietary diversity and subsequent mortality in the First National Health and Nutrition Examination Survey Epidemiologic Follow-up Study. American Journal of Clinical Nutrition. 1993; 57(3):434-40) and (Farhangi MA, Jahangir L. Dietary diversity score is associated with cardiovascular risk factors and serum adiponectin concentrations in patients with metabolic syndrome. BMC Cardiovascular Disorders. 2018; 18(1)). These authors work with dietary diversity, rather than diet diversity. On the other hand, we considered that to be counted as a “consumer” of any of the food group’s categories, a respondent needed to consume for at least 1 day one-half serving as defined by the Spanish Food Pyramid quantity criteria. The Spanish Food Pyramid does not include a minimum adequate intake for non-healthy foods. Other authors, have also measured dietary diversity for only the recommended foods (Kant AK, Thompson FE. Measures of overall diet quality from a food frequency questionnaire: National Health Interview Survey, 1992. Nutrition Research. 1997; 17(9):1443-56). Moreover, we decided to exclude non-recommended food groups (such as high salt, sugar or saturated fatty acids) because their negative or positive effect depend on the amount ingested. In fact, the more important question is the percentage of total energy supplied by these food groups and all our analyses are adjusted by total energy. We have added a comment on this to the main text (line 204-207).

9.      Line 236: The authors provide three categories of physical activity (less active, moderately active, or active); and alcohol intake. Yet, neither the variables nor the categorisation were defined. A similar comment applies to other demographic variables. However, these are more easily understood and thus do not necessarily need a definition (e.g. civil status, smoker).

We have completed the methods section (“Assessment of non-dietary variables”) with a brief description of the categorization used for physical activity. Currently in the main text appears the following information “Individuals were classified based on their level of physical activity using a validated Spanish version of the Minnesota questionnaire: less active (< 4 MET), moderately active (4–5.5 MET), and active (≥ 6 MET) physical activity level” (line 251-253).

We have also included information on alcohol intake. The information that appears now is: “alcohol intake (measured as a continuous variable and expressed as intake in g/d)” (line 249-250).

10.   Line 266: The authors write that participants in the top DDS quartile had lower WC (not surprising), and lower education (surprising!). A comment on the latter would be warranted in the discussion section.

In table 1 we have showed raw data for baseline characteristics of our sample of study. As we showed, the percentage of women is greater in quartiles 3 and 4 of DDS. This could explain the increased percentage of older aged persons (women participating in the study were older than men), the lower WC and also the lower educational level of participants in the top DDS quartile. According to the unexpected result regarding lower education, we have included in the Discussion section the following phrase: “The distribution of our population reflects the social and demographic characteristics of the Spanish population born in the 40’s-60’s. In that context women did not have the opportunity to reach high levels of formal education. As the percentage of women is greater in the top quartile of DDS, this could be an attributable factor that explains that subjects with a higher DDS have a lower educational level”, this explanation has been included in line 395-399.

11.   Whereas nearly all of the analyses presented are statistically significant, one does wonder about the meaningfulness about some of the differences. For example, the difference in mean age between the quartiles is significant (p<0.001), but corresponds to a mean difference of approximately 1.5 years. Considering the age range of the population is 55-75 years, what does this really mean? Is this difference relevant?

Differences shown in Table 1 and Table 2 of the new version of the manuscript should be interpreted in the light of the large sample size. However, we agree with the reviewer that results should not only be interpreted based on the p-value but also on the size of the effect. The effect in some cases is not meaningful. We thank the reviewer for commenting on this and we have included some thoughts about it in the first paragraph of the Results section (line 285-287).

12.   Line 277: What does "religious and single status" refer to? This was not described in the methods.

In methods we described the civil status as a categorical variable with four possible alternatives (married, widowed, divorced/separated and others). In the “others” category, we have included single participants and those who are priests or nuns, and that category was named as “religious”. We have now included this information in the methods section, where we have specified that the “others” category includes these two alternatives (lines 246-248).

13.   Table 2: Similar to the statistical differences reported in Table 1, one wonders about the meaningfulness of the statistical differences in some of the nutrients upon stratification by quartile.

Thank you for the suggestion. We have modified the results section accordingly (see lines 318– 319).

14.   Table 3: Interpretation of this table would have been easier if the authors had provided a n for each gender-age group.

As per the suggestions made by another reviewer, we have moved the table 3 included in the main text to supplementary tables (now supplementary table 1). Nevertheless, we have included the sample size according to quartiles of DDS in order to not overload the table. The information about the percentage of participants stratified by sex and age according to DDS quartiles is presented in table 1.

15.   Table 4: Please consider the presentation of this table. At present, the lengthy variable names are repeated several times. The table could be substantially edited for clarity of presentation.

Thank you very much for this comment. We have edited the table in order to enhance the clarity of result`s presentation. A new version, is now available in the main text.

16.   Table 5: Please provide labels for what are presumably OR and 95%CI.

We have included additional information in the title and table`s legend, specifying the following information “Values are presented as OR and 95%CI for inadequate intake of micronutrients as categorical variable according to Total DDS and food`s group diversity” (line 362 and line 363-364).  

17.   Table 5: Some of the point estimates change considerably with partial- and full adjustments. Consider if this is the the result of overestimation. If so, how could this be addressed?

We think that changes from the unadjusted model to the adjusted models are due to confounding. As shown in table 1, baseline characteristics of participants are different across quartiles of DDS, and therefore, they are potential confounding factors. To clarify Table 5, and because the unadjusted model is clearly confounded, we have decided to remove model 1 from Table 5. The new version of Table 5 is now more readable and less misleading.

18.   Table 5: For Total DDS, why was no Model 1 presented?

As you can see in the Methodology section and at legend of the table, the Model 1 was the unadjusted model, meanwhile the model 2 (Adjusted by energy) and model 3 (moreover adjusted by other variables) where the adjusted models.  As we described in methodology, Total DDS was adjusted by total energy intake.  Therefore we could not add Model 1 for total DDS. However, as we mentioned above, to clarify Table 5, we have decided to remove model 1 from Table 5. The new version of the table is now available in page 11-12 of the manuscript.

19.   Lines 332-334: In the summary sentence of the discussion, the authors note that "special attention" should be paid to several groups with lower DDS. Yet, it is not clear to which quartiles the authors are referring. For example, amongst the widow/widowers, the biggest difference is seen between Q2 and Q3. For other groups, such as alcohol drinkers, interpretation is hindered by the lack of definition of variables and the categorisation of the variables.

Thank you for the suggestion. We have rephrased the sentence, so that it now makes more sense in the light of results presented in Table 2. You can read the new text at lines 354-357.

20.   Lines 365-369: If these studies cannot be compared to your study, why mention them in detail, or at all?

In the discussion (lines 365-369) we compare our results with those from 2 studies with similar objectives than ours. However, their results shows differences with ours. Therefore we discuss the reasons for this lack of similarity. In order to make it easier for the reader, we have made some small modifications in that paragraph (lines 369-374).

21. Line 415: Do the authors mean N, rather than n?

Effectively, when we wrote n we refer to sample size. In order to refer to “sample size” we have written “n” throughout the manuscript.

Round 2

Reviewer 3 Report

Thank you to the authors and English language editor for a much improved manuscript. At this time, I have only 2 comments for consideration.

The authors write "Furthermore, the distribution of our population reflects the social and demographic characteristics of the Spanish population born in the 40’s-60’s. In that context, women  did not have the opportunity to reach high levels of formal education." Please consider writing 1940s-1960. As well, please revise to "women had limited opportunity to pursue high levels of formal education."

2. The last sentence of the conclusion "Health policies should focus on the promotion of a healthy varied diet, specifically promoting the intake of a variety of vegetables and fruits among population groups with lower DDS such as men, smokers or widow(er)s people" is, in fact, not a conclusion. Although this message is important, it would be better placed earlier in the discussion.

Author Response

REVIEWER 3- ROUND 2

Thank you to the authors and English language editor for a much improved manuscript. At this time, I have only 2 comments for consideration.

1.       The authors write "Furthermore, the distribution of our population reflects the social and demographic characteristics of the Spanish population born in the 40’s-60’s. In that context, women  did not have the opportunity to reach high levels of formal education." Please consider writing 1940s-1960. As well, please revise to "women had limited opportunity to pursue high levels of formal education."

-We really apreciate the time that you have spent reading and reviewing this article, as well as the positive comments that you have made about the reviewed version of the manuscript. We agree with both comments. We have rewritten the sentence, currently appears as: Furthermore, the distribution of our population reflects the social and demographic characteristics of the Spanish population born in the 1940s-1960. In that context, women had limited  opportunity to pursue high levels of formal education”. This information appears at lines 353-355  

2.       The last sentence of the conclusion "Health policies should focus on the promotion of a healthy varied diet, specifically promoting the intake of a variety of vegetables and fruits among population groups with lower DDS such as men, smokers or widow(er)s people" is, in fact, not a conclusion. Although this message is important, it would be better placed earlier in the discussion.

-We really appreciate your suggestion and totally agree. We have moved this information. Currently appears at lines 426-428 of the discussion section.
